# Multiplanar Semicircular New-Generation Implant System Developed for Proximal Femur Periprosthetic Fractures: A Biomechanical Study

**DOI:** 10.3390/medicina61010110

**Published:** 2025-01-14

**Authors:** Ahmet Burak Satılmış, Ahmet Ülker, Zafer Uzunay, Tolgahan Cengiz, Abdurrahim Temiz, Mustafa Yaşar, Tansel Mutlu, Uygar Daşar

**Affiliations:** 1Department of Orthopedics and Traumatology, Taşköprü State Hospital, Kastamonu 37400, Turkey; tolgahancengiz@hotmail.com; 2Department of Orthopedics and Traumatology, Mersin University, Mersin 33110, Turkey; drahmetulker@gmail.com; 3Department of Orthopedics and Traumatology, Medicalpark Adana Hospital, Adana 01060, Turkey; dr.zuzunay@gmail.com; 4Department of Industrial Design Engineering, Karabük University, Karabük 78050, Turkey; abdurrahimtemiz@karabuk.edu.tr (A.T.); myasar@karabuk.edu.tr (M.Y.); 5Department of Orthopedics and Traumatology, Medicalpark Gebze Hospital, Kocaeli 41400, Turkey; tanselmutlu@yahoo.com; 6Department of Orthopedics and Traumatology, Karabük University, Karabük 78050, Turkey; udasar@yahoo.com

**Keywords:** new generation, implant, periprosthetic fractures, fracture fixation, biomechanics, hip arthroplasty

## Abstract

*Background and Objectives***:** The study aimed to evaluate a newly designed semicircular implant for the fixation of Vancouver Type B1 periprosthetic femoral fractures (PFFs) in total hip arthroplasty (THA) patients. To determine its strength and clinical applicability, the new implant was compared biomechanically with conventional fixation methods, such as lateral locking plate fixation and a plate combined with cerclage wires. *Materials and Methods*: Fifteen synthetic femur models were used in this biomechanical study. A Vancouver Type B1 periprosthetic fracture was simulated by osteotomy 5 mm distal to the femoral stem. The models were divided into three groups: Group I (lateral locking plate fixation), Group II (lateral locking plate with cerclage wires), and Group III (new semicircular implant system). All fixation methods were subjected to axial loading, lateral bending, and torsional force testing using an MTS biomechanical testing device. Failure load and displacement were measured to assess stability. *Results*: The semicircular implant (Group III) demonstrated a significantly higher failure load (778.8 ± 74.089 N) compared to the lateral plate (Group I: 467 ± 68.165 N) and plate with cerclage wires (Group II: 652.4 ± 65.474 N; *p* < 0.001). The new implant also exhibited superior stability under axial, lateral bending, and torsional forces. The failure load for Group III was more robust, with fractures occurring at the screw level rather than plate or screw detachment. *Conclusions*: Compared to traditional fixation methods, the newly designed semicircular implant demonstrated superior biomechanical performance in stabilizing Vancouver Type B1 periprosthetic femoral fractures. It withstood higher physiological loads, offered better structural stability, and could be an alternative to existing fixation systems in clinical practice. Further studies, including cadaveric and in vivo trials, are recommended to confirm these results and assess the long-term clinical outcomes.

## 1. Introduction

According to the latest annual report, the incidence of total hip replacement (total hip arthroplasty, THA) surgery was reported as 360 per 100,000 in individuals aged 40 and over. The prevalence of THA in the total US population was found to be 0.83% in 2010, meaning approximately 2.5 million people underwent hip replacement surgery that year [1]. Demand for primary THAs in the US is projected to increase by more than half a million by 2030 [2]. Factors such as increasing life expectancy and using cementless fixation increase the incidence of periprosthetic femoral fractures (PFFs). Abdel et al. reported a 1.7% fracture rate for all primary THAs operated on at the Mayo Clinic in the USA between 1969 and 2011 [3].

A complication of hip replacement surgery is a periprosthetic femoral fracture (PFF) around or distal to the hip replacement. These fractures can occur during prosthesis placement (intraoperative) or after surgery (postoperative). Most postoperative fractures occur after a fall from the same level and are most common in patients between the ages of 70 and 90 [4]. These fractures have high complication and mortality rates. Many patients do not return to their previous activity levels despite the fracture healing [5]. Patient satisfaction after PFF treatment is low, and costs are relatively high [6,7].

The Vancouver classification system is widely used in the classification of PFFs [8]. This classification considers the fracture’s location, the stem’s stability, and the bone’s quality around the stem. The Vancouver classification divides fractures into three types. In Type A, the fracture is an avulsion of the greater or lesser trochanter. Type B fractures occur around or just below the stem, while fractures well below the stem tip are classified as Type C. Type B fractures are divided into three subcategories: B1, B2, and B3. The Vancouver classification system is a helpful tool for selecting treatments. Vancouver Type A fractures are usually managed with conservative treatment, and partial weight bearing is recommended for separations of less than 2 cm; surgical intervention may be required for greater separation or if problems persist despite treatment. Vancouver Type B1 fractures are treated with open reduction and internal fixation, while B2 and B3 fractures are usually treated with stem revision, although alternative treatment methods are also considered in patients with poor general conditions. Surgical fixation methods are preferred for Type C fractures [9].

The number of studies on treatment methods for periprosthetic femur fractures developing after total hip arthroplasty is increasing daily. Biomechanical studies have gained momentum to find the optimum fixation method. Fixation methods created with plate–screw systems, cerclage wires, cables, bone grafts, revision prostheses, and their combinations have been investigated in many studies with clinical, radiological, biomechanical, or finite element analysis methods [10,11,12]. In this study, we designed a semicircular implant similar to the Ilizarov system as an alternative to the lateral plate–screw system in biomechanical studies on Vancouver Type B1 periprosthetic fractures and compared it biomechanically with current treatment methods. Thus, we aimed to compare the new implant with other fixation methods biomechanically and evaluate its clinical applicability in Vancouver Type B1 periprosthetic femur fractures.

## 2. Materials and Method

The study aimed to obtain a more biomechanically robust implant design than the existing fixation methods. The previous fixation methods were considered in the implant design and their deficiencies were evaluated. In this way, the developed implant consisted of three large clamps and provided proper containment according to the bone’s dimensions. There are three M6 screw holes in each clamp; two locked cortical screws are used in the proximal and distal clamps. The implant is supported by three rods that reduce the load on the fractured area and transfer the load to the other parts of the bone. The clamps prevent axial shifts in the fractured part of the bone and ensure that most of the force is transmitted through the rods (Figure 1).

In this study, with the number determined by power analysis, 15 right femur models (Synbone AG, Malans, Switzerland, model 2230) specially produced from polyurethane foam, having an internal structure resembling trabecular bone structure and a hard outer shell surrounding this structure imitating cortical bone, were used. The models were produced for orthopedic surgical education and biomechanical studies and have been safely used in many studies previously published in the literature. With these models used, the variability between experimental samples in terms of femoral anatomy was minimized, and using a single type of model allowed us to eliminate structural differences that may occur in implant fixation and aim to obtain homogeneous study groups for each fixation method. In the preliminary study conducted to find the femoral component that best fit the femur bone models, a neck osteotomy was performed with an electric saw 1.5 cm proximal to the lesser trochanter at a 45-degree angle with the femoral shaft. After the medulla was carved and prepared, it was rasped with the largest number 4 rasp. X-ray checks were performed to ensure proper placement, and a Zimmer uncemented standard straight size four femoral stem (Alloclassic^®^ Zweymüller^®^ Schaft SL, Zimmer GmbH, Winterthur, Switzerland) was implanted following the manufacturer’s instructions and using the appropriate material.

Then, 15 bone models with femoral stems were divided into three groups, with five bones in each group, and Vancouver Type B1 fracture was simulated with osteotomy. While creating this fracture model, a horizontal cut was made proximally, a 45° cut was made distally, and an osteotomy was made 5 mm distal to the stem tip. In designing the study, the medial defect in the experimental model was used to evaluate implant performance under harsh conditions. Future studies should include models without a medial defect and alternative fixation methods such as medial double-plate fixation to provide more clinically relevant comparisons. For the fixation of Group I, only a 10-hole plate was applied laterally. Starting from the upper end of the plate, four proximal and four distal holes were drilled with a drill bit. After the measurement, fixation was made to these holes with 14 mm monocortical locking screws proximally and 38 mm bicortical locking screws distally (Figure 2). For the fixation of Group II, only a 10-hole plate and cerclage wires were applied laterally. Starting from the upper end of the plate, the 2nd and 4th holes and 4 distal holes were drilled with a drill bit. These holes were fixed with 14 mm monocortical screws and two 2.0 mm cerclage wires proximally, and 38 mm bicortical locking screws distally (Figure 3). In Group III, the implant group, the design and the three-dimensional drawing, which was made in our clinic, were produced on a five-axis CNS bench. This newly produced implant was used for the fixation of Group III. The fixation was made with two 16 mm monocortical screws in the proximal clamp and 38 mm bicortical locking screws in the distal clamp. A 2.0 mm cerclage wire was used in these clamps (Figure 4).

Biomechanical tests were performed at the Iron and Steel Institute, Margem Biomechanics Laboratory, using a test device (MTS Systems Landmark, Eden Prairie, MN, USA) that could apply axial forces to biomaterials and measure biomechanical changes in materials. Axial insufficiency loading tests were applied to all models in the study groups created using different fixation methods. For axial loading, the direction of the resultant force on the femoral head was ensured to be 20 degrees valgus to mimic the single-leg stance phase of walking. A metal device fixed to the lower end of the device was used for all models. A +0 CoCr head was placed on the femoral stems, and the force was transferred to the femoral stem from an acetabulum-like socket corresponding to the head. The femoral models were fixed using the same metal device by ensuring the appropriate configuration of the MTS device before each experiment (Figure 5). Force and displacement values were recorded with the MTS Axial LoadCell (MTS Systems Landmark, Eden Prairie, MN, USA). The biomechanical compression test was performed at a 5 mm/min measurement range, and axial loading continued until failure was developed. If a decrease was observed after the peak point in the force–displacement graph and this decrease continued during long-term follow-up, it was accepted that failure had developed due to the deterioration of the implant structure and the system’s structure, and the test was terminated. In this test, the movement (displacement) in the fracture line, the highest force (failure load) that caused the failure, and the displacement at this force (failure displacement) values were recorded.

Statistical data were evaluated using Statistical Package for the Social Sciences (SPSS) for Windows version 20.0. Descriptive statistics for categorical variables were presented as numbers and percentages, and, for numerical variables, as mean ± standard deviation and minimum–maximum values. In the analysis of numerical data, compliance with normal distribution was examined using the “Kolmogorov–Smirnov” and “Shapiro–Wilk” tests, and since the only numerical variable, maximum Newton data, showed normal distribution, the mean difference between the three groups was examined using the “One-Way ANOVA” test. In cases with a significant difference, pairwise group analyses were analyzed using the “Tukey” test. Data were examined at a 95% confidence level, and tests were considered significant if the *p*-value was less than 0.05. The study was conducted by the Non-Interventional Clinical Research Ethics Committee of a tertiary university hospital with the decision dated 28.03.2018 and numbered 4/31.

## 3. Results

The load passing through the femoral shaft during the single-leg stance phase of a 70 kg individual can be approximated as 1372 N to 2058 N. However, these values may vary depending on the individual’s stance position, walking style, and other factors [13]. In the study, finite element analysis was performed, and the system, which was subjected to a fixed load of 10 Newtons (N), was loaded starting from 500 N for axial tension and increasing by 250 N up to 6000 N. In Group III, the amount of deformation and von Mises stress values that occurred in the system when maximum forces were applied with the axial loading test were calculated. Accordingly, a deformation of 7.81 mm occurred in the fracture line under 6000 N force. It was seen that the maximum stress during this maximum loading was in the proximal screw of the implant.

The torsional force seen on the femoral shaft of a 70 kg individual varies between 100 and 400 N [14]. In the lateral bending and torsion test, loading was applied starting from 0 N and increasing by 250 N up to 750 N. The amount of deformation and the von Mises stress values that occurred in the system when these forces were used were calculated. Accordingly, under the torsional force of 750 N, a deformation of 4.44 mm occurred in the fracture line, and the maximum stress was seen in the distal clamp of the implant. When lateral bending was applied, a deformation of 3.49 mm occurred in the fracture line, and the maximum stress was seen in the distal clamp of the implant. The axial load and lateral bending–torsion test results obtained with finite element analysis show that the newly developed implant is quite useful in PFFs.

In compression tests, the highest force (failure load) causing failure on the implant and the displacement at this force (failure displacement) values were calculated directly from the slope of the load–displacement curve recorded by the MTS device. Accordingly, the average axial failure load of the models after axial loading was found to be 467 ± 68.165 N in Group I, 652.4 ± 65.474 N in Group II, and 778.8 ± 74.089 N in Group III (Table 1). These results show that Group III was statistically superior to the other groups for axial failure load and failure displacement (*p* < 0.001).

The analysis of which two groups caused this significant difference was made, and both groups were compared among themselves; while no statistically significant difference was found between Group 1 and Group 2 (*p* = 0.063), the differences between Group 2 and Group 3 (*p* < 0.001) and Group 1 and Group 3 (*p* < 0.001) were found to be significant. The average maximum newton value of Group 2 was numerically higher than the average maximum newton value of Group 1. The average maximum newton value of Group 3 was statistically significantly higher than the average maximum newton value of Groups 1 and 2. The failure load test of the models in Group I was in the form of the separation of the proximal screws from the bone together with the plate in all models. The failure load test of the models in Group II was in the form of a transverse fracture from the distal of the plate in all models. The failure load test of the models in Group III was in the form of a transverse fracture from the screw level located distal to the implant. The stability of the prosthesis was not impaired in any of the models. All these results show that the new implant system applied in Group III was biomechanically more potent than the classical fixation methods used in Groups I and II in PFFs.

## 4. Discussion

With the continuing aging of the modern population, the number of primary hip arthroplasties implanted continues to increase. Complications such as periprosthetic fractures also occur more frequently following this increase. Today, up to 4% of patients who have undergone primary hip arthroplasty will experience at least one periprosthetic fracture during their lifetime [15], the most common etiology being a fall on the prosthetic hip. The treatment of periprosthetic fractures depends primarily on the stability of the arthroplasty. Osteosynthetic procedures are usually applied to fractures with stable arthroplasty, while fractures with a loosened endoprosthesis usually require revision arthroplasty. The therapeutic strategy should always be related to the fracture location and bone quality. The aim is to reestablish the correct length, axis, and bone substance and, thus, the patient’s movement as quickly as possible with stable osteosynthesis or revision arthroplasty.

The Vancouver classification system is widely used in PFFs. Vancouver B fractures constitute the vast majority of periprosthetic proximal femur fractures. In Vancouver B1 fractures (stable primary implant), osteosynthesis retaining arthroplasty is generally the preferred treatment method. However, there is no consensus on the reduction type of the fracture (open–closed), the fixation material to be used, the characteristics of the plate (locking–nonlocking), whether or not cortical strut allograft will be used, its position if used, and how proximal fixation will be performed when using the plate. Although many clinical and biomechanical studies have been conducted on these, research is still ongoing for the most stable fixation method. The variety of techniques, implants, and combinations means no “gold standard” treatment exists. In our study, a Vancouver Type B1 fracture model, frequently encountered in PFFs, was created, and the applicability and biomechanical strength of the implant were investigated.

In the studies conducted, fractures can be created differently in femur models. In their research, Zdero et al. performed an osteotomy at a 45° angle from the distal of the femoral stem with a 5 mm gap in the fracture line to create a periprosthetic fracture [16]. Dennis et al. performed fixation with the same osteotomy without creating a gap in the fracture line [17]. In their study, Wahnert et al. performed an osteotomy 5 mm distal to the femoral stem, with the proximal being horizontal and the distal being at a 45° angle [18]. In their study on periprosthetic femur fractures, Wilson et al. performed horizontal osteotomy from the distal part of the femoral stem and fixed the fracture line without leaving any gap [19]. In our study, a horizontal cut was made proximally and a 45° cut was made distally to create a fragmented periprosthetic femur fracture model, and an osteotomy was made 5 mm distal to the stem tip. Again, when the literature is reviewed, it is seen that there are differences between the angle that the femur makes with the ground when performing the axial loading test, and there is no consensus. In these studies, the femur was placed in the valgus between 5° and 25° to mimic the single-footed phase of walking. When the walking cycle and the loads on the femur are examined, it is seen that the loads on the femur in one cycle are between 12.5° and 21° [20]. In our study, we applied 20° valgus loading, which is between these values.

Vancouver Type B1 fractures are usually treated with open reduction and internal fixation [21]. Locking compression plates have been used as an alternative method of fracture fixation. Plate-locking screws allow for single-cortex fixation at multiple points. Biomechanical studies have shown that these plates are more stable than nonlocking devices [22]. Recent biomechanical studies have shown that locking plates are superior to traditional cable–plate fixation in stabilizing periprosthetic femoral fractures in axial loading and torsional forces [23,24]. Locking screws provide better fixation, mainly when single-cortex screws are used in the proximal fracture segment of osteoporotic bone. Conventional screws have the advantage of being angled anterior and posterior to the femoral stem. Unlike the locking screw–plate system, they allow compression in simple transverse or oblique fractures. Cables increase fixation in the proximal segment and allow the addition of strut cortical grafts to increase stability and provide a mechanical and potential biological advantage in osteoporotic bone.

Lenz et al. compared the biomechanical performance of different cable and wire cerclage configurations, concluding that cable cerclages provided a higher fixation strength compared to single-wire cerclages but were similar to double-wound wire cerclages [25]. Sandhu et al. performed open reduction and Dall–Miles plating in 20 patients with periprosthetic femoral fractures with stable femoral components. All fractures healed without fixation failure within 1–4 years postoperatively. However, two varus Type B1 fractures subsequently developed, and both cases were treated with a plate fixed with cables alone. Based on these results, the authors recommend that plate fixation with cables alone should be avoided because of the torsional instability of the construct [26]. In a biomechanical study, Dennis et al. showed that plate constructs with proximal single-cortex screws and distal bicortical screws or proximal single-cortex screws, proximal cables, and distal bicortical screws were significantly more stable in axial compression, lateral bending, and torsional loading than a plate system with cables alone [17].

Zdero et al., in a biomechanical in vitro study conducted with Sawbone in 2008, compared osteosynthetic treatment options for Vancouver B1 and C periprosthetic fractures. They created models with a locking plate and locking screw in group 1, locking plate and proximal locking screw–cable in group 2, locking plate and proximal locking screw–cable in group 3, and a locking plate and anterior strut graft in group 4. They compared these models’ axial, rotational, bending stiffness, and axial insufficiency loading values. As a result, they showed that the axial, rotational, and bending stiffness of the group with a locking plate and strut graft was significantly higher than the other groups [16]. In a human cadaver study, Wähnert et al. compared a fixed-angle locking attachment plate (LAP^®^, Depuy Synthes^®^, Solothurn, Switzerland) with a variable-angle non-contact bridging plate (NCB^®^, Zimmer GmbH, Winterthur, Switzerland) in the Vancouver B1 fracture model. In their study, the non-contact bridging plate system withstood higher loads than the fixed-angle locking plate. The non-contact bridging plate may be the preferred option when high stability and load capacity are needed immediately after surgery [27]. In a biomechanical study on synthetic bone, Lenz et al. demonstrated that the locking attachment plate construct provided improved mechanical stability and strength compared to the cerclage construct [28].

As a result of all these studies, we can say that the most stable fixation method in Vancouver Type B1 periprosthetic femoral fractures is open reduction, a locking plate combined with a cortical strut graft, and proximal fixation using unicortical screw and cable. However, it is challenging to obtain strut allografts today. Although they provide biomechanically stable fixation, they cause many complications, such as nonunion, fracture, and infection, when applied in treating PFFs. Considering these disadvantages, our new-generation implant system can be an alternative method by providing stable fixation under physiological loads in all three axes.

The positive aspects of the study are that there is no previous study in the literature with a semicircular fixation system; it is an experimental study; homogeneous groups are formed using materials with similar properties; comparisons are made between groups, and the results apply to the clinic. The findings support the biomechanical superiority of the semicircular implant. However, our study has some limitations. Our study was conducted in vitro. There was no soft tissue cover on the models. Compared to the in vivo environment, fixation methods were performed very quickly. In addition, this study could not simulate the surrounding muscle and ligament structures that may change mechanical properties. Although the mechanical properties and anatomy of the models are similar to human bones, the bones used are not real bones and do not have the osteoporotic bone feature frequently seen in periprosthetic fractures. Using a medial defect model poses an additional limitation, as such defects are uncommon in clinical Vancouver Type B1 fractures. Future studies will incorporate models without medial defects, compare performance with medial dual-plate fixation systems, and evaluate long-term in vivo outcomes.

## 5. Conclusions

There is no consensus on the treatment of PFFs. The new semicircular implant we designed for Vancouver Type B1 periprosthetic femoral fractures is a biomechanically superior method. It can withstand physiological loads, has a low failure rate under pathological loads, and does not damage the prosthesis or implant. Biomechanical and computer-aided studies should be conducted to support these results by applying the implant to a cadaver. Animal experiments should be performed to evaluate the clinical results, and the results should be supported.

## Figures and Tables

**Figure 1 medicina-61-00110-f001:**
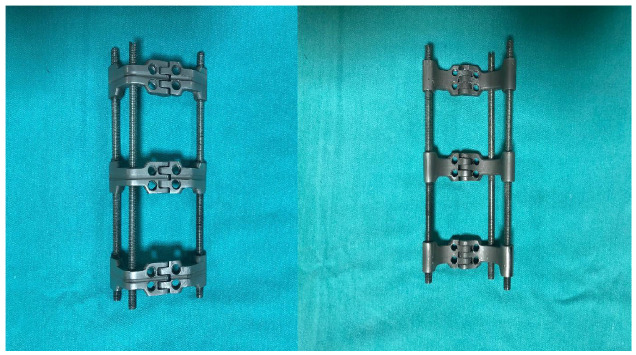
Designed new-generation implant system.

**Figure 2 medicina-61-00110-f002:**
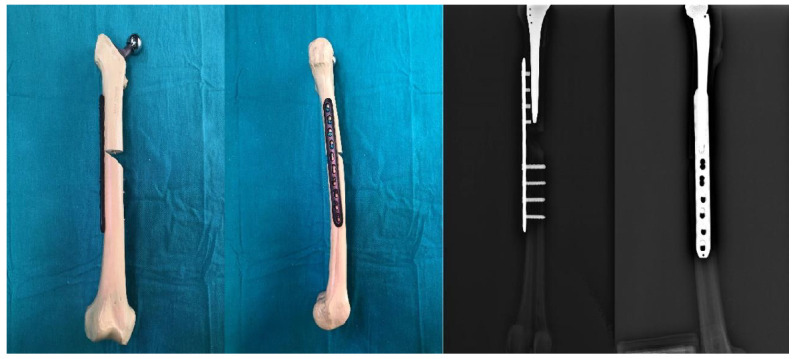
Preparation and radiological appearance of Group I.

**Figure 3 medicina-61-00110-f003:**
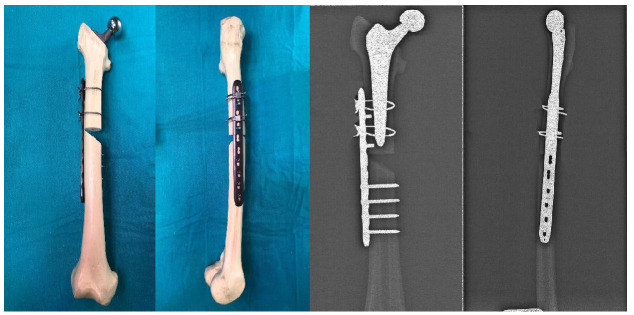
Preparation and radiological appearance of Group II.

**Figure 4 medicina-61-00110-f004:**
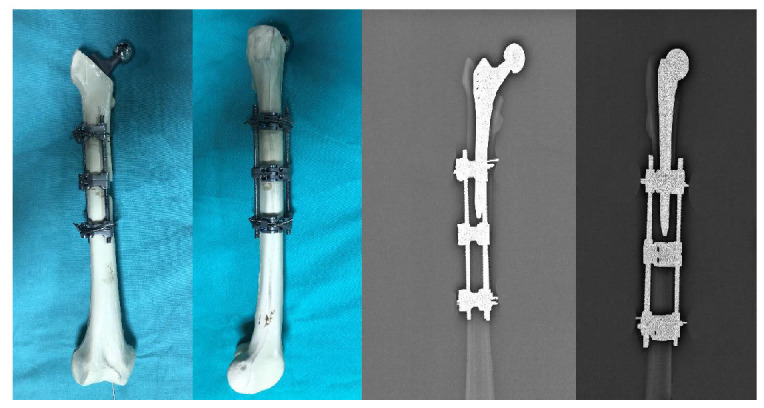
Preparation and radiological appearance of Group III.

**Figure 5 medicina-61-00110-f005:**
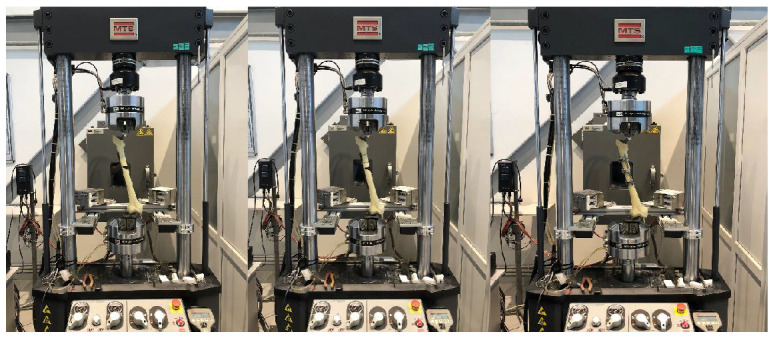
Views of the groups inside the test device while testing.

**Table 1 medicina-61-00110-t001:** Results of axial load and failure load tests.

Group	N	Mean	Standard Deviation	Minimum Displacement	Maximum Displacement
1	5	467	68.165	396	555
2	5	652.40	65.474	538	698
3	5	778.80	74.089	688	882
Total	15	632.73	142.270	396	882

## Data Availability

All data can be transmitted to the journal if requested.

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
