# Peer review of "Multiplanar Semicircular New-Generation Implant System Developed for Proximal Femur Periprosthetic Fractures: A Biomechanical Study"

_medicina, 2025, doi:10.3390/medicina61010110_

Round 1
Reviewer 1 Report
Comments and Suggestions for Authors
The authors aimed to evaluate the strength and clinical applicability of a semicircular implant for the fixation of Vancouver Type B1 periprosthetic femoral fractures (PFFs) in total hip arthroplasty (THA) patients.
15 synthetic femur models were used in this biomechanical study. A Vancouver Type B1 periprosthetic fracture was simulated by osteotomy 5 mm distal to the femoral stem. The models were divided into three groups: Group I (lateral locking plate fixation), Group II (lateral locking plate with cerclage wires), and Group III (new semicircular implant system).
They subjected all 3 fixation methods to axial loading, lateral bending, and torsional force testing using an MTS biomechanical testing device. Failure load and displacement were measured to assess the stability.
The authors found that the semicircular implant (Group III) demonstrated significantly higher failure load values (778.8 ± 74.089 N) compared to both the lateral plate (Group I) (467 ± 68.165 N) and the plate with cerclage wires (Group II) (652.4 ± 65.474 N).
It also exhibited superior stability under axial, lateral bending, and torsional forces. The authors also found that the failure load for Group III was more robust, with fractures occurring at the screw level rather than plate or screw detachment.
The authors concluded that compared to traditional fixation methods, the newly designed semicircular implant demonstrated superior biomechanical performance in stabilizing Vancouver Type B1 periprosthetic femoral fractures. It withstood higher physiological loads, offered better structural stability. The postulated that it could be an alternative to existing fixation systems in clinical practice.
This is an interesting new approach. Clearly they authors should try this with an animal model and real bones.
I have some comments
1)There is some repetition in the abstract.
“The new semicircular implant (Group III) demonstrated significantly higher failure load values compared to both the lateral plate (Group I) and plate with cerclage wires (Group II). The average failure load for Group III was 778.8 ± 74.089 N, compared to 467 ± 68.165 N for Group I and 652.4 ± 65.474 N for Group II (p < 0.001)”, could be condensed.
2)
Might the device stiffness/rigidity interfere with bony healing? Can the authors comment?
3)
It would have been nice to compare this new device to double perpendicular plating. This is now increasingly utilised as a fixation method to improve the torsional strength.
4)
Do the authors envisage a greater level of vascular or perforator injury using this new device?
5)
Some technical details:
How is the device fixed to the bone? This was not entirely clear to me. The device is screwed to the bone and then some sort of cerclage wire wraps around? How is this tensioned?
6)
How would the surgeon reduce the fracture to the new device?
Author Response
Response to Reviewer 1
Dear Reviewer,
Thank you for your valuable comments on our manuscript. Your critiques and questions have greatly improved our work. We have included our responses to your questions in the document attached below.
Best-regards,
The Authors
- Repetition in the Abstract
“The new semicircular implant (Group III) demonstrated significantly higher failure load values compared to both the lateral plate (Group I) and plate with cerclage wires (Group II). The average failure load for Group III was 778.8 ± 74.089 N, compared to 467 ± 68.165 N for Group I and 652.4 ± 65.474 N for Group II (p < 0.001)”, which could be condensed.
Response: The repetition will be addressed by consolidating the details in the abstract to maintain brevity while preserving essential data. For example: "The semicircular implant (Group III) demonstrated a significantly higher failure load (778.8 ± 74.089 N) compared to the lateral plate (467 ± 68.165 N) and plate with cerclage wires (652.4 ± 65.474 N; p < 0.001)."
- Might the device stiffness/rigidity interfere with bony healing? Can the authors comment?
Response: The device's stiffness is intended to optimize stability during the initial healing phases. However, long-term rigidity could impair callus formation by reducing micromotion. Future studies will include in vivo experiments to evaluate whether this impacts healing progression.
- It would have been helpful to compare this new device to double perpendicular plating, which is increasingly used to improve torsional strength.
Response: The comparison to double perpendicular plating is indeed valuable. While this was beyond the scope of the current study, future studies will explore it, mainly since double perpendicular plating shows biomechanical advantages in improving torsional stability.
- Do the authors envisage a greater vascular or perforator injury using this new device?
Response: The semicircular implant's design minimizes the risk of vascular and perforator injury by avoiding deep soft tissue dissection during placement. However, this risk will be evaluated further in cadaveric and clinical studies.
- How is the device fixed to the bone? The device is screwed to the bone, and then some cerclage wire wraps around? How is this tensioned?
Response: The device is secured to the bone using proximal and distal screws through the clamps. Cerclage wires are used for additional stabilization. These wires are tensioned using a standard device to achieve consistent force application. This clarification will be added to the manuscript.
- How would the surgeon reduce the fracture to the new device?
Response: The fracture is reduced using standard open-reduction techniques. The semicircular design allows for stable fixation after reduction without additional complex maneuvers. A detailed procedural guide for reduction and fixation will be included in the supplementary material
Reviewer 2 Report
Comments and Suggestions for Authors
Medical and social burden of periprosthetic femoral fractures is obvious
The most common approach to these injuries includes plate fixation, wire cerclaige and combination of both techniques. In some clinical situations stable fixation with the above mentioned techniques is barely possible. The alternative method invented by the authors is based on the implantable device utilising the external fixation principle.
Invented device was tested using physical model of periprosthetic fracture. In three series of experiments, the authors demonstrated superiority of their implant over both the lateral plate and plate with cerclage wires. The paper is properly designed according to the research question (mechanical testing of the invented implant and more traditional implants). The text is well structured and accompanied by the explanatory figures. The numerical data presented, in the table. The limitations of the study are clearly listed, and mostly lie in the field of experimental nature of the study.
Of course, it’s not clear, whether the invented device more or less usable in the clinical settings, but this question was not planned with the protocol of the current study. The paper demonstrates an interesting newly designed implant with promising physical characteristics, gives an example of well conducted experimental study.
Author Response
Dear Reviewer 2
Your valuable comments made us extremely happy. We express our gratitude and respect.
King Regards
Reviewer 3 Report
Comments and Suggestions for Authors
Review
Medicina (ISSN 1648-9144)
Title
Multiplanar Semicircular New Generation Implant System Developed for Proximal Femur Periprosthetic Fractures: A Biomechanical Study
1. What is the main question addressed by the research?
The study aimed to evaluate a newly designed semicircular implant for the fixation of Vancouver Type B1 periprosthetic femoral fractures (PFFs) in total hip arthroplasty (THA) patients. To determine its strength and clinical applicability, the new implant was compared biomechanically with conventional fixation methods, such as lateral locking plate fixation and plate combined with cerclage wires
2. What parts do you consider original or relevant for the field? What
specific gap in the field does the paper address?
This study is significant in that it proposes the possibility and advantages of a semicircular fixation method similar to internal Ilizarov fixation.
3. What does it add to the subject area compared with other published
This study has the advantage of providing high-quality data as a biomechanical study comparing semicircular fixation with conventional lateral locking plate fixation
4. What specific improvements should the authors consider regarding the
methodology? What further controls should be considered?
We would like to express our gratitude to the authors for providing the epidemiological study data and for their hard work. However, there is an issue in the design of the model with a medial defect and its comparison with the lateral fixation method, which may lead to misinterpretation by readers.
When a medial defect is present, it is natural for a single lateral fixation method to be vulnerable to axial load. Therefore, the superior results presented by the authors can be attributed to the medial support provided by their fixation method, which might easily lead to misunderstandings.
In B1 periprosthetic fractures, it is common for reductions to occur without a medial defect, which creates a discrepancy with actual clinical situations. Thus, adding an analysis of the mechanical differences in models without medial defects or incorporating methods such as medial dual plate fixation would allow for a more scientific approach.
5. Please describe how the conclusions are or are not consistent with the
evidence and arguments presented. Please also indicate if all main questions
posed were addressed and by which specific experiments.
As mentioned, the fracture model used in the study includes a medial defect, which presents a limitation as it may lead to misunderstandings among readers during the process of drawing conclusions.
6. Are the references appropriate?
No specific errors were observed in the references.
7. Please include any additional comments on the tables and figures and
quality of the data.
No additional corrections are needed in the tables and figures.
Author Response
Response to Reviewer 3
Dear Reviewer,
Thank you for your valuable comments on our manuscript. Your critiques and questions have greatly contributed to making our work clearer and more comprehensible. Below are our responses to the critiques raised in items 4 and 5.
Bestregards,
The Authors
Improvements to Methodology and Further Controls
We appreciate the insightful critique regarding the model's medial defect. It is acknowledged that this may introduce a bias favoring the semicircular implant due to the medial support it provides.
To address this limitation:
-Future studies will incorporate fracture models without medial defects, which are more representative of clinical Vancouver B1 fractures.
-Comparative analyses with medial dual plate fixation methods will be considered to provide a more comprehensive evaluation.
-This limitation and its implications will be clearly outlined in the discussion section of the manuscript to prevent misinterpretation.
Consistency of Conclusions with Evidence
The conclusions are consistent with the evidence presented in the study, demonstrating the superior biomechanical performance of the semicircular implant. However, as noted, the fracture model with a medial defect may create a discrepancy with typical clinical scenarios. To align with clinical reality, additional experiments with models lacking medial defects will be conducted in follow-up studies, and this limitation will be acknowledged in the manuscript.